# FAdV-4 Promotes Expression of Multiple Cytokines and Inhibits the Proliferation of aHEV in LMH Cells

**DOI:** 10.3390/v15102072

**Published:** 2023-10-10

**Authors:** Lidan Hou, Wei Wang, Zengna Chi, Yawen Zhang, Zhong Zou, Peng Zhao

**Affiliations:** 1China Institute of Veterinary Drug Control, Beijing 100081, China; houruif@163.com; 2Zhaoyuan Center for Disease Control and Prevention, Yantai 265400, China; 15054515053@163.com; 3College of Animal Science and Veterinary Medicine, Shandong Agricultural University, Taian 271018, China; 13625321478@163.com (Z.C.); 18754808250@163.com (Y.Z.); 4Hubei Jiangxia Laboratory, Wuhan 430200, China

**Keywords:** fowl adenovirus (FAdV), avian hepatitis E virus (aHEV), genetic analysis, co-infection

## Abstract

Single or mixed infections of multiple pathogens such as avian hepatitis E virus (aHEV) and avian leukosis virus subgroup J (ALV-J) have been detected in numerous laying hens with severe liver injury in China. Thus, aHEV and immunosuppressive viruses are speculated to cause co-infections. In this study, co-infection with aHEV and fowl adenovirus (FAdV) was confirmed by nested RT-PCR and recombinase-aided amplification combined with gene sequencing in two flocks with severe liver injury. Subsequently, the two reference strains, aHEV and FAdV-4, were inoculated into LMH cells to identify their co-infection potential. Confocal microscopy revealed aHEV and FAdV-4 co-infected LMH cells. In addition, the replication dynamics of aHEV and FAdV-4 along with the expression levels of immuno-cytokines were measured. The results indicated colocalization of aHEV and FAdV-4 and inhibition of viral replication in LMH cells. The transcription levels of MDA5, Mx, OASL, and IFN-α were significantly upregulated in LMH cells, whereas those of immune-related factors induced by FAdV-4 were downregulated upon FAdV-4 and aHEV co-infection. These results confirmed the co-infection of aHEV and FAdV-4 in vitro and prompted the antagonistic pathogenic effects of FAdV-4 and aHEV, thereby providing novel insights into the counterbalancing effects of these viruses.

## 1. Introduction

Since 1991, hepatitis E virus (HEV) has been detected in chickens in Canada and the United States [1]. The main clinical symptoms include ovarian degeneration, abdominal hemorrhage, swelling of the liver and spleen, and reduced egg production [2]. Molecular epidemiological investigations have shown that while avian HEV (aHEV) is prevalent in chickens in the United States and Spain [3,4], most chickens have a subclinical course without obvious symptoms. Recently, cases of aHEV have been reported in Europe, China, Hungary, South Korea, and other regions, but none of these strains have caused severe diseases [5,6,7,8].

However, from 2016 onwards, an epidemic associated with aHEV infection has occurred in several farms’ hens in China, most of which suffer from liver damage, causing huge economic losses to the poultry industry [9,10]. Studies have demonstrated that although the fatality rate of aHEV is low, co-infection with other immunosuppressive viruses, such as MDV or avian leukosis virus subgroup J (ALV-J), can aggravate clinical symptoms, leading to a decrease in the laying rate or an increase in the mortality of chickens [11,12,13,14]. 

Recently, fowl adenovirus (FAdV)-4 infections have been described as co-infections involving pathogens such as infectious bursal disease virus, avian reoviruses, Marek’s disease virus, and chicken infectious anemia virus (CIAV) [1,1,15,16]. The increasing spread of these viruses within the poultry sector represents a major concern regarding the economic consequences of co-infection. Upon viral infection, the expression level of immune-related factors determines the antiviral ability of the host. Studies have revealed that cytokines such as IL-1, IL-6, OASL, IFN-α, and Mx are crucial regulators of antiviral processes. Therefore, quantitative analysis of their expression could aid in further understanding of the interplay between the viruses and the host immune response. 

Previously, co-infection with aHEV and FAdV-4 was identified in three Hy-line brown laying hens with hepatomegaly and splenomegaly; however, the underlying mechanism of co-infection and its effects on viral replication remain unclear. Therefore, in this study, we compared and analyzed the viral proliferation titers as well as transcription levels of immune-related factors in aHEV and FAdV-4 single-infected and co-infected groups.

## 2. Materials and Methods

### 2.1. Cells

Leghorn male hepatoma (LMH) cell line (ATCC CRL-2117) cells were maintained in Dulbecco’s modified Eagle’s medium supplemented with 10% fetal bovine serum (Gibco, San Diego, CA, USA), 100 U/mL penicillin, and 100 μg/mL streptomycin at 37 °C in a humidified atmosphere of 5% CO_2_.

### 2.2. Infected Chickens and Their Sample Collection

On a Hy-line brown commercial laying hen breeding farm (no. LN) in Liaoning Province, China, chickens began to exhibit white combs and increased mortality at 270 days of age, with a mortality rate of approximately 0.2% per day. Necropsy revealed that the livers of dead chickens were enlarged and brittle, and the abdominal cavities of some chickens were filled with blood. Another Hy-line brown parental layer farm (No. HB) in Hebei Province, China, had concentrated deaths at approximately 80 days of age with a weekly mortality rate of up to 3%. Furthermore, necropsy revealed big liver and spleen (BLS) disease in all dead chickens. In total, 500 serum samples were randomly collected from individuals in the two groups. Commercial enzyme-linked immunosorbent assay (ELISA) kits (IDEXX Laboratories Inc., Beijing, China) were used to detect antibodies against ALV-J, ALV-A, REV, and CIAV according to the manufacturer’s instructions. All results were negative for the tested antibodies. In contrast, it was speculated that the chickens were co-infected with FAdV-I and aHEV, as tests for both FAdV-I and aHEV using commercial ELISA kits (BioCheck, South San Francisco, CA, USA) were positive. The positivity rates of FAdV-I and aHEV antibodies in farm HB were 62% and 30%, respectively, and the positivity rates of FAdV-I and aHEV antibodies in farm LN were 48% and 22%, respectively.

### 2.3. Detection of FAdV-I Nucleic Acid in Liver Samples of Infected Poultry

Thirty liver samples were obtained from dead chickens, and DNA was extracted from the liver tissue using a DNA extraction kit (OMEGA, Bio-Tek, Norcross, GA, USA) according to the manufacturer’s instructions. In this study, the nucleic acids of FAdV-I were detected using the fluorescence recombinase-aided amplification (RAA) and recombinase-aided amplification–side flow chromatography strip (RAA-LFD) detection methods (China National Patent No. 202110889250.5). Primers and probes for fluorescent RAA and RAA-LFD detection were designed and synthesized from the conserved domains of 12 serotype reference strains of FAdV-I [17]. Sensitivity tests for the standard plasmid demonstrated that the minimum detectable concentration of fluorescent RAA and RAA-LFD was 100 copies/µL. Based on the known sequence of FAdV-I, primers for amplifying its fiber gene were designed and synthesized. Fiber-F (5′-GTTCCCGCCTCGTTATTG-3′) and Fiber-R (5′-AGTGGGACAGACTGATGG-3′) were used to amplify the fiber gene of the isolated strain. All primers were synthesized by Sangon Biotech Co., Ltd. (Shanghai, China). The amplified PCR products were subjected to 1% agarose gel electrophoresis and purified using an agarose gel DNA purification kit (OMEGA, USA), according to the manufacturer’s protocol. The corresponding fragments were confirmed by Sanger sequencing. 

### 2.4. aHEV Detection and Sequence Analysis in Liver Samples of Diseased Chickens

RNA was extracted using commercial kits (OMEGA, Bio-Tek, Norcross, GA, USA), according to the manufacturer’s instructions. For first-strand complementary DNA (cDNA) synthesis, total RNA was reverse transcribed using random primers. Based on a previously published aHEV nucleic acid detection method [15], aHEV was detected in dead chickens using a nested reverse transcription polymerase chain reaction (RT-PCR). The sequences of the external primers were ORF2/F-1/SD (5′-TCGCCT(C)GGTAAT(C)ACA(T)AATGC-3′) and *ORF2*/R-1/SD (5′-GCGTTC (G) CCG (C) ACAGGT (C) CGGCC-3′) in the first round of PCR, and the length of the target fragment was 278 bp. The primer sequences for internal use were ORF2/F-2/SD (5′-ACA (T) AATGCT (C) AGGGTCACCCG-3′) and ORF2/R-2/SD (5′-ATGTACTGA (G) CCA (G) CTG(C) GCCGC-3′) in the second round, and the length of the target fragment was 242 bp. Primers used in this study were synthesized by Sangon Biotech Co., Ltd. (Shanghai, China). For the positive samples detected by Nest-PCR, the ORF2 genes of aHEV-positive samples were amplified using amplification primers based on a previous study [18]. The amplified PCR products were subjected to 1% agarose gel electrophoresis, purified using an agarose gel DNA purification kit (OMEGA), according to the manufacturer’s protocol, and sent to Sangon Biotech Co., Ltd. for sequencing.

### 2.5. Detailed Experimental Protocol for Infection of aHEV and FAdV-I in LMH Cells 

To verify whether aHEV and FAdV-I can co-infect cells in vitro, LMH cells were used. As shown in Figure 1, the LMH cells were infected/co-infected with the aHEV wild-type strain YT (GenBank accession number: MZ736614.1) and serum type 4 FAdV-I wild-type strain GY (GenBank accession number: 2532406). Group A was inoculated with FAdV-4 GY only, whereas group B was inoculated with aHEV-YT. Group G was co-infected with a mixture of FAdV-4 GY and aHEV-YT at the same dosages as in groups A and B. Group C was inoculated with FAdV-4 GY at the same time as group A and inoculated with the aHEV-YT strain 24 h later, and group E was the HEV single-infected group inoculated with aHEV-YT at the same time as Group C. Group D was inoculated with aHEV-YT at the same time as group B and then inoculated with the FAdV-4 GY strain 24 h later, while group F was the single-infected group inoculated with the FAdV-4 GY strain at the same time as group D. LMH cells cultured in DMEM were used as the control group, and all the experiments with co-infected groups were set in quadruplicates. At 48 h, 72 h, and 96 h after inoculation, 200 µL of cell culture supernatant was collected and frozen at −80 °C for further examination. After removing the cell culture supernatant, an equal volume of the cell maintenance solution was added.

### 2.6. Confirmation of aHEV and FAdV-I Co-Infection in LMH Cells

To confirm the co-infection of LMH cells with aHEV and FAdV-I, IFA was performed following the standard procedure. For IFA, LHM cells were cultured on coverslips in six-well plates infected with aHEV and/or FAdV-I. Polyclonal antibodies (pAbs) (1:200) against aHEV capsid protein (previously prepared in our laboratory) [19] or monoclonal antibodies (mAbs) (1:500) against FAdV-I penton protein (previously prepared in our laboratory) were used as primary antibodies. LMH cells were stained by indirect immunofluorescence using rabbit anti-HEV serum and mAb against FAdV-I penton as the primary antibodies, and the colocalization of HEV and FAdV-I in LMH cells was observed using a laser confocal microscope (Carl Zeiss, Jena, Germany).

### 2.7. The Effect of aHEV and FAdV-I Co-Infection on Cytokine Expression Levels in LMH Cells 

After DNA extraction from the cell culture supernatant, qPCR for FAdV-I was performed as previously described [14]. After RNA extraction, qRT-PCR for aHEV was performed as described previously [20]. The replication levels of aHEV and FAdV-I were measured in single infection and co-infection groups. Twelve pairs of primers were synthesized as described previously (Table 1) [21,22]. These primers were used to detect the levels of interleukin 1 (IL-1), interleukin 6 (IL-6), interleukin 8 (IL-8), interleukin 18 (IL-18), Toll-like receptor 3 (TLR3), Toll-like receptor 7 (TLR7), melanoma differentiation-related gene 5 (MDA5), mitochondrial antiviral signaling proteins (MAVS), and other proteins involved in innate immune response in LMH cells. The CT values of each cytokine in each sample obtained by qPCR were subtracted from the CT values of β-actin, which served as an internal reference. Then, the average ΔCT of each treatment group was subtracted from the ΔCT of the blank group to obtain ΔΔCT, and the fold change (=2^ ^(−ΔΔCT))^ was calculated. If the difference was greater than 1, the expression of this cytokine was considered to be upregulated compared to that in the untreated group. On the contrary, if the difference was between 0 and 1, the expression of this cytokine was downregulated compared to that in the untreated group. 

### 2.8. Statistical Analysis

All experiments were reproducible and performed in triplicates. Statistical analyses were conducted using a two-way ANOVA test to compare the differences between two groups using GraphPad Prism (version 5.0; GraphPad Software, La Jolla, CA, USA). Statistical significance was set at *p* < 0.05.

## 3. Results

### 3.1. Identification of aHEV and FAdV-I Co-Infection in Liver Samples of Hy-Line Brown Dead Chickens

Postmortem findings revealed that all dead chickens showed typical symptoms of liver enlargement, and several chickens had liver rupture. RAA technology was used to detect the viral infection in liver samples of dead chickens and control nucleic acids. Amplification curves were exhibited in 10–15 min for the dead chicken samples, whereas no amplification was observed for the negative control nucleic acid, indicating that liver samples of dead chickens in both groups were infected with FAdV-I (Figure 2). The results of the RAA-LFD technique further confirmed this conclusion. Four liver samples from the two flocks of dead chickens showed significant color bands in the control and test regions of the LFD strip (Figure 3), suggesting the presence of FAdV-I nucleic acids; however, the specific serotype of FAdV-I could not be determined. The fiber gene of the liver samples was sequenced and compared using BLAST in the NCBI database. The results indicated that the fiber gene in both flocks maintained the highest homology with serum type 4 FAdV-I and was in the same branch (Figure 4); that is, both flocks were infected with serum type 4 FAdV-I. The amplification results of the ORF2 partial sequences showed that liver samples from dead chickens in the two flocks were positive for aHEV infection. Sequence alignment of the ORF2 sequences using BLAST in NCBI suggested that the ORF2 sequences maintained the highest homology with aHEV and were in the same branch (Figure 5). In summary, the above results indicated that co-infection with serum type 4 FAdV-I and aHEV was detected in both flocks of chickens.

### 3.2. Co-Localization of aHEV and FAdV-I Infection in LMH Cells

Studies have described the use of avian-derived cell models in studies of FAdV-4 and aHEV, including the use of LMH and DF1 cell lines. IFA was performed using rabbit anti-aHEV-ORF2 protein serum as the primary antibody and tetramethyl rhodamine isothiocyanate (TRITC)-labeled sheep anti-rabbit IgG as the secondary antibody. The results showed that the cytoplasm was red in both the aHEV single-infected group and the aHEV and FAdV-I co-infected group. Hence, aHEV successfully infected LMH cells. The FAdV-I universal monoclonal antibody mAB-Penton-6^#^ was used as the primary antibody, and fluorescein isothiocyanate (FITC)-labeled sheep anti-mouse IgG was used as the secondary antibody. The results revealed that the cytoplasm in both the FAdV-I single-infected group and FAdV-I and aHEV co-infected groups showed typical green fluorescence staining. Hence, LMH cells were successfully infected with FAdV-I. Confocal microscopy was used to visualize the co-localization of FAdV-I and aHEV. It demonstrated that multiple cells were simultaneously infected with FAdV-I and aHEV (Figure 6). Taken together, we established a co-infection cell model by inoculating LMH cells with FAdV-4 and aHEV.

### 3.3. Effects of aHEV and FAdV-I on Each Other’s Replication upon Co-Infection in LMH Cells 

Next, we observed the effects of aHEV and FAdV-I co-infection in LMH cells on their replication. It is known that due to its limited replication capacity, aHEV enters the exponential stage gradually compared to other viruses after entering the plateau stage; it is stable at a relatively low level. In a preliminary experiment, it was observed multiple times that the viral replication of different groups was significantly different at 72 h; thus, the replication of the two viruses at 72 h was compared. As shown in Figure 7, the copy number of FAdV-4 GY was detected upon simultaneous inoculation with aHEV-YT (Figure 7A), upon inoculation with aHEV-YT 24 h before FAdV-4 GY inoculation (Figure 7B), or 24 h after FAdV-4 GY inoculation (Figure 7C). Overall, co-infection with the three inoculation modes inhibited the replication of FAdV-4 GY to varying degrees. When detecting the copy number of aHEV-YT, infection with both aHEV-YT and FAdV-4 GY occurred simultaneously; the average copy number of the co-infected group inoculated with FAdV-4 GY was 126.9. Conversely, the average copy number of the group inoculated with aHEV-YT alone was 787.7, which is, six times higher than that of the co-infected group (Figure 7D). Regardless of FAdV-4 GY inoculation 24 h before (Figure 7E) or after aHEV-YT inoculation (Figure 7F), replication of aHEV-YT was significantly inhibited in the process of co-infection compared to that in the single-infected group. In conclusion, co-infection of LMH cells with aHEV-YT and FAdV-4 GY inhibited viral replication.

### 3.4. Effect of Co-Infection of HEV and FAdV-4 on Cytokines

As shown in Figure 8, after 48 h of infection, LMH cells infected with FAdV-4 GY (group A) showed significant upregulation of the transcription levels of most immuno-cytokines (IL18, TLR7, IFN-α, and IFN-β excepted), such as OASL, Mx, MDA5, and MAVS. In contrast, the aHEV-YT single-infected group (Group B) did not show significant upregulation in the transcription levels of the aforementioned genes. Meanwhile, aHEV-YT and FAdV-4 GY co-infection (group G) led to a decrease in the transcription levels of the above factors that were induced by FAdV-4 GY. At 96 h after infection, the transcription levels of immune-related factors such as OASL, Mx, and IFN-α in the FadV-4 GY single-infected (group A) and co-infected groups (group G) were maintained at high levels, while the above factors were not significantly upregulated in the aHEV-YT single-infected group (group B). However, co-infection with aHEV-YT also decreased the transcription levels of these factors induced by FAdV-4 GY (Figure 9). LMH cells were first infected with FAdV-4 GY and then further infected with aHEV-YT, it was evident that aHEV-YT could still reduce the transcription levels of immuno-cytokines related to FadV infection. For example, 48 h after infection, the transcription levels of OASL, Mx, MDA5, and MAVS were upregulated in the FAdV-4 GY single-infected group (group A) and the group subjected first to FAdV-4 GY infection with aHEV-YT and then infection with aHEV-YT (group C) after culturing with LMH cells. However, the transcription levels in the co-infected group were significantly lower than those in the FAdV-4 GY single-infected group. At 96 h after infection, the transcription levels of OASL and Mx showed the same pattern as above; that is, co-infection with aHEV-YT downregulated the transcription levels of most of the immune-related factors detected in this study that were otherwise induced by FAdV-4 GY single infection.

## 4. Discussion

Since 2016, hepatic rupture hemorrhagic syndrome (HRHS) with hepatosplenomegaly has occurred in several chicken flocks in China. However, the underlying cause remains unclear. aHEV has been identified several times in related flocks. It is considered the main pathogen in chicken liver and spleen (BLS) disease as well as HRHS. These diseases mainly affect laying hens and broiler breeders, resulting in hemoperitoneum, fatty amyloidosis of the liver, BLS, and HRHS. However, aHEV nucleic acids have also been detected in healthy flocks, raising the suspicion that aHEV may not be the sole agent of these clinical symptoms [9,10,23,24,25]. Co-infection with aHEV and ALV-J has been reported in broilers and layers [11,12]. In this study, co-infection with aHEV and FAdV-I was detected in two Hy-line brown laying flocks. This is the first report to demonstrate co-infection with aHEV and FAdV-I under natural conditions.

To observe the co-infection of aHEV and FAdV-I and their interaction*s* in vitro, we built a cell model using LMH cells. Co-infection with aHEV and FAdV-I in the same cells was observed using laser confocal microscopy, indicating that both aHEV and FAdV-I could infect LHM cells. Furthermore, the dynamics of the replication status of aHEV and FAdV-I during co-infection were detected by qRT-PCR. Interestingly, we found that that aHEV-YT and FAdV-4 GY inhibited each other’s replication during co-infection in LMH cells.

IFN and other innate immune-related factors constitute the first line of host defense against pathogenic infections. They stimulate the production of IFN stimulators by the host via IFN receptors, thereby inhibiting viral replication and spread. Various cytokines, such as type I IFN, play important roles in the replication and pathogenicity of aHEV and FAdV-4. In view of this, we further observed and compared the expression of multiple immune-related factors upon single and co-infections with aHEV and FAdV-4. The results showed that LMH cells infected with FAdV-4 showed high expression of multiple immune-related factors in the short term, which is consistent with previous reports [26]. For example, second-generation sequencing has been used to perform transcriptome studies on LMH cells infected with FAdV-4. Sequencing data obtained at three time points after viral infection identified a total of 7000 genes with significant differences in transcription [27]. Further analysis revealed that these differentially expressed genes were involved in a series of biological processes, including metabolism, innate immunity, inflammatory responses, and signal transduction. Among them, the Toll-like receptor, JAK-STAT, MAPK, and cytokine-cytokine receptor interaction signaling pathways related to host cell innate immunity and signal transduction were affected. In addition, TLR2A, TLR3, TLR5, My D88, IL-12B, IL-12RB2, IL-5RA, IL-18, IL-21R, CCL20, CXCL14, and a series of differentially expressed cytokine and cytokine receptor genes were identified. These differentially expressed genes may be closely related to the inflammatory response induced by FAdV-4 infection. Previous studies in geese detected differences in FAdV-4 load and immune-related gene expression using RT-qPCR. In addition, dynamic changes in mRNA transcription levels of Toll-like receptors in FAdV-4-infected LMH cells showed that 10 types of TLRs were upregulated to varying degrees, especially in the latter period of infection (72–120 h) (*p* < 0.05 or *p* < 0.01). Among these, the levels of TLR1a, TLR1b, TLR4, TLR7, and TLR21 increased significantly. This study and previous reports suggest that the inflammatory response and cell damage caused by FAdV-4 infection are closely associated with the expression of multiple immune-related cytokines.

Unlike the strong immune response induced by FAdV-4 infection, aHEV does not induce high expression levels of immune-related factors. For instance, the transcript levels of IFN-α and IFN-β were lower in aHEV infected cells. Compared to FAdV-Ⅰ single infection, the transcription levels of IFN-α and IFN-β along with MDA5, MAVS, and Mx decreased significantly after co-infection, suggesting a series of chain reactions [28]. As an intracytoplasmic nucleic acid receptor, MDA5 is highly similar to retinoic acid-induced expression gene I (RIG-I) and specifically recognizes dsRNA after binding to pathogen-associated molecular patterns (PAMPs). MDA5 interacts with the adapter protein CARD homologously through its own CARD domain and further binds to MAVS to interact with it so that the RIG-I receptor (RLR) can be relocated to the inner membrane. Additionally, the TRAF2/TRAF6-activated IKK kinase complex can be recruited to activate the transcription factor NF-κB. In contrast, TRAF3 and TBK1 are recruited to promote the phosphorylation and activation of IRF3. The activated transcription factors NF-κB and IRF3 enter the nucleus and work together to promote the transcription and expression of the type I IFN gene, whereas the Mx protein is an antiviral protein induced by type I IFN. It has been speculated that aHEV mainly inhibits the expression of type I IFN and Mx proteins by inhibiting the transcription of MDA5.

In conclusion, this study confirmed the co-infection of aHEV and FAdV-I in Hy-line brown laying hens in China and suggested that it might be the cause of morbidity in these two flocks. Further in vitro studies showed that co-infection with aHEV and FAdV-I could occur in LMH cells. Additionally, aHEV inhibited the expression of IFN and other cytokines, whereas co-infection with FAdV-I and aHEV significantly inhibited the mutual replication of each other. The results of this study provide further insight into the effect of co-infection with FAdV-I and reference information for the analysis of the cause and possible pathogenesis of hepatosplenomegaly and rupture syndrome prevalent in laying hens China.

## 5. Conclusions

Our study reports the co-infection of aHEV and FAdV-4 in two flocks of Chinese laying hens. It also shows that these viruses can infect the LMH cell line individually as well as simultaneously in vitro. The counterbalancing replication regulation of both the viruses upon co-infection has been indicated by monitoring the expression levels of innate immune response proteins and cytokines. Altogether, these results provide a novel insight into the antagonistic pathogenic effects of FAdV-4 and aHEV on each other. 

## Figures and Tables

**Figure 1 viruses-15-02072-f001:**
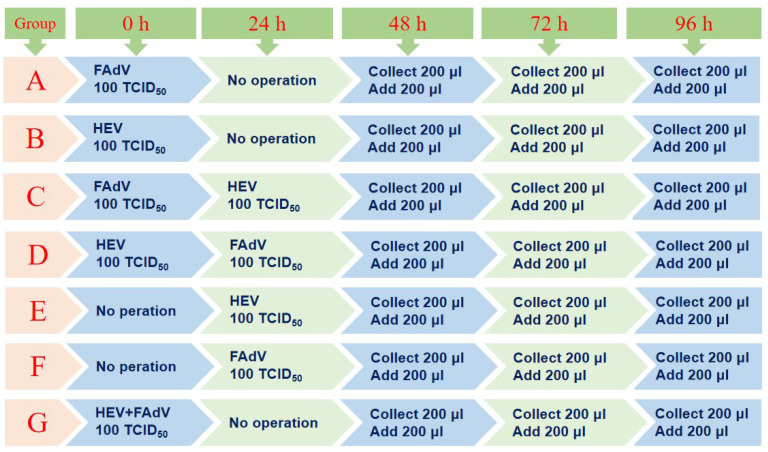
Experimental design of artificial infection of HEV and FAdV-I in LMH cells. A: inoculated with FAdV-4 GY; B: inoculated with aHEV-YT; C: inoculated with FAdV-4 GY and then inoculated with the aHEV-YT; D: inoculated with aHEV-YT and then inoculated with the FAdV-4 GY; E: single-infected group inoculated with aHEV-YT at the same time as Group C; F: single-infected group inoculated with the FAdV-4 GY strain at the same time as group D; G: co-infected with a mixture of FAdV-4 GY and aHEV-YT at the same time.

**Figure 2 viruses-15-02072-f002:**
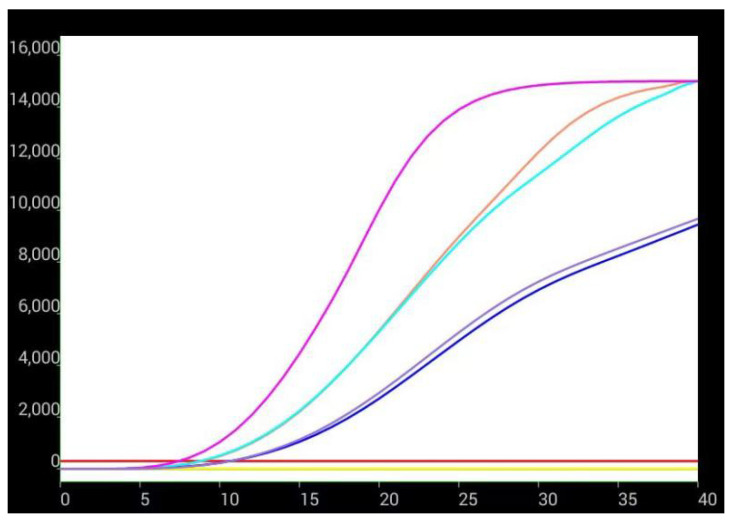
FAdV-I nucleic acid detection in dead chicken livers by RAA technique. X: reaction times (minutes); Y:fluorescence (F). The different coloured curves represent nucleic acid samples.

**Figure 3 viruses-15-02072-f003:**
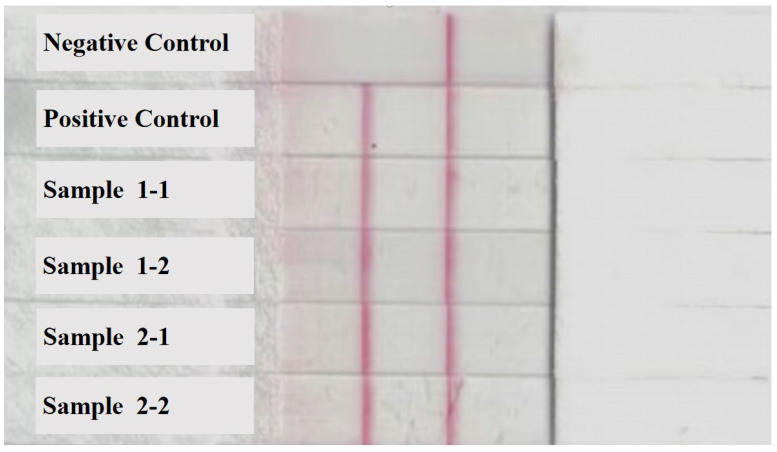
FAdV-I nucleic acid detection in dead chicken livers by RAA-LFD technique.

**Figure 4 viruses-15-02072-f004:**
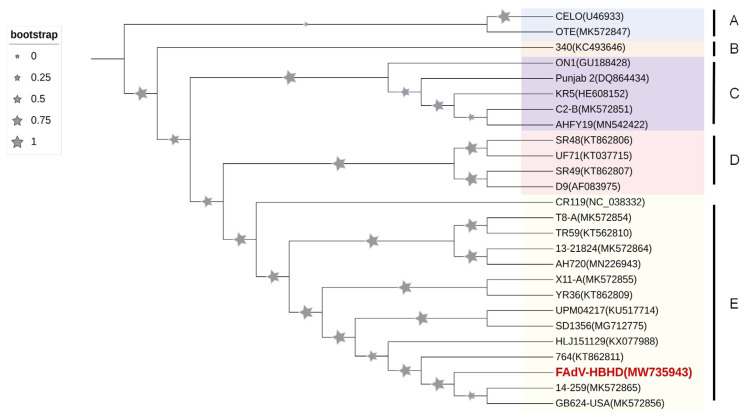
Phylogenetic trees constructed based on the nucleotide sequences of fiber of FAdV-I in clinical samples.

**Figure 5 viruses-15-02072-f005:**
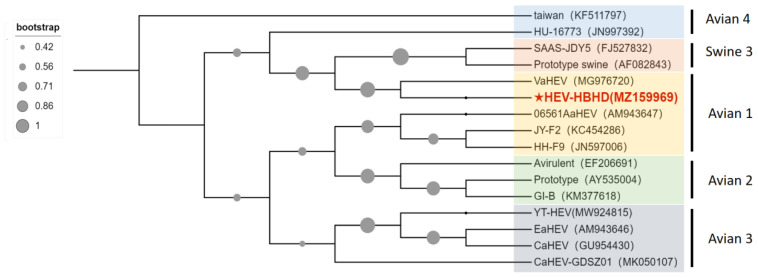
Phylogenetic trees constructed based on the nucleotide sequences of ORF2 of aHEV in clinical samples.

**Figure 6 viruses-15-02072-f006:**
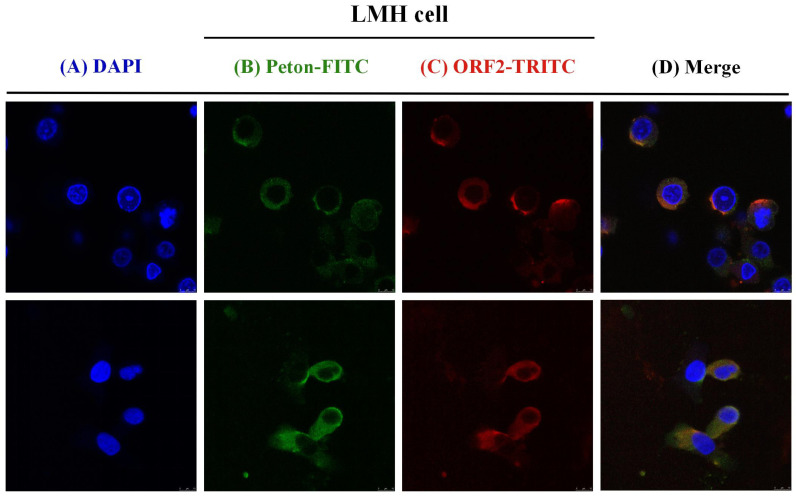
Co-localization of aHEV-YT and FAdV-4 GY in LMH cells by confocal laser microscopy (**A**) DAPI staining; (**B**) FITC-labeled Penton mAb; (**C**) TRITC-labeled ORF2 pAb; (**D**) merged.

**Figure 7 viruses-15-02072-f007:**
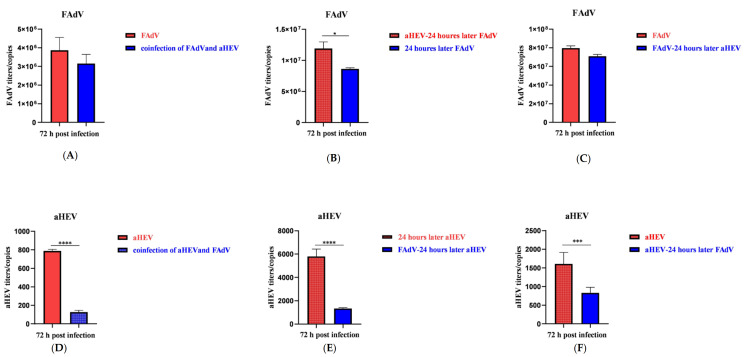
Copy numbers of FAdV-4 (**A**–**C**) and aHEV (**D**–**F**) under different conditions as indicated in single- or co-infected LMH cells. * *p* < 0.05, *** *p* < 0.001 and **** *p* < 0.0001.

**Figure 8 viruses-15-02072-f008:**
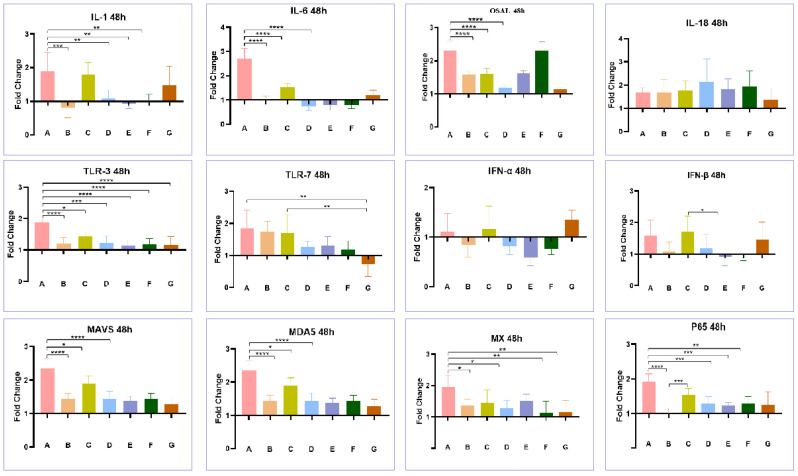
Transcription levels of immune-related genes in LMH cells with single or co-infection by aHEV and FAdV-4 at 48 hpi. The examined immune-related factors include IL-1, IL6, OSAL, IL-18, TLR-3, TLR-7, IFN-α, IFN-β, MAVS, MDA5, Mx, and P65. * *p* < 0.05, ** *p* < 0.01, *** *p* < 0.001 and **** *p* < 0.0001.

**Figure 9 viruses-15-02072-f009:**
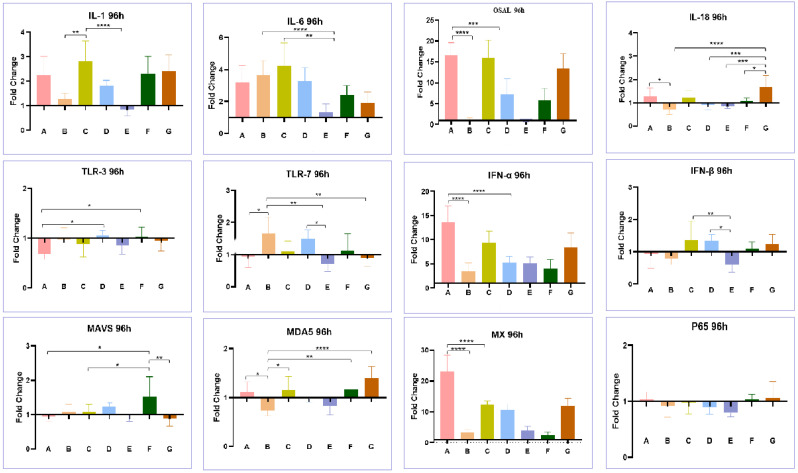
Transcription levels of immune-related factors in LMH cells with single or co-infection by aHEV and FAdV-4 at 96 hpi. The examined immune-related factors include IL-1, IL6, OSAL, IL-18, TLR-3, TLR-7, IFN-α, IFN-β, MAVS, MDA5, Mx, and P65. * *p* < 0.05, ** *p* < 0.01, *** *p* < 0.001 and **** *p* < 0.0001.

**Table 1 viruses-15-02072-t001:** Primers used for detection of virus replication and immune-related factors.

Gene	Direction	Sequences (5′–3′)
FAdV	Forward	ATGGCKCAGATGGCYAAGG
	Reverse	ATGGCKCAGATGGCYAAGG
aHEV	Forward	AATGTGCTGCGGGGTGTCAA
	Reverse	CATCTGGTACCGTGCGAGTA
p65	Forward	CCACAACACAATGCGCTCTG
	Reverse	AACTCAGCGGCGTCGATG
Mx	Forward	CAGCTCCAGAATGCATCAGA
	Reverse	GGCAATTCCAGGAAGATCAA
TLR3	Forward	GCAACACTTCATTGAATAGCCTTGAT
	Reverse	GCCAAACAGATTTCCAATTGCATGT
TLR7	Forward	AAGTCCCGGTATGTTCAGCT
	Reverse	GGACAGGGTATTGTTCATAGC
MDA5	Forward	CAGCCAGTTGCCCTCGCCTCA
	Reverse	AACAGCTCCCTTGCACCGTCT
MAVS	Forward	CCTGACTCAAACAAGGGAAG
	Reverse	AATCAGAGCGATGCCAACAG
IL-1	Forward	ATGACCAAACTGCTGCGGAG
	Reverse	AGGTGACGGGCTCAAAAACC
IL-6	Forward	GACGAGGAGAAATGCCTGACG
	Reverse	CGAGTCTGGGATGACCACTTC
OASL	Forward	GAGATGGAGGTCCTGGTGAA
	Reverse	CCAGCTCCTTGGTCTCGTAG
IL-18	Forward	GAGGTGAAATCTGGCAGTGG
	Reverse	GAATGTCTTTGGGAACTTCTCC
IFN-α	Forward	TACGGCATCCTGCTGCTCAC
	Reverse	AGAGAAGGTGGCATCCTGGG
IFN-β	Forward	GCCCACACACTCCAAAACACTG
	Reverse	TGATGCTGAGGTGAGCGTTG
β-Actin	Forward	CCCACCTGAGCGCAAGTACT
	Reverse	AAGCATTTGCGGTGGACAAT

## Data Availability

All data are presented along with the article.

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
