# Peer review of "FAdV-4 Promotes Expression of Multiple Cytokines and Inhibits the Proliferation of aHEV in LMH Cells"

_viruses, 2023, doi:10.3390/v15102072_

Round 1
Reviewer 1 Report
please refer to the acctached fils.

It would be better if the manuscirpt could be improved by english-native writer.
Author Response
Thanks for your comments, please see attached.

Reviewer 2 Report
1. Line 23-24: “These results confirmed …….and prompted the an tagonistic pathogenic effects of FAdV-4 on aHEV, thereby….. ” should be adjusted to “These results confirmed…..and prompted the an tagonistic pathogenic effects of FAdV-4 and aHEV, thereby…..”.
2. Line 153: “other proteins involved in innate immune response in LMH cell supernatants” should be adjusted to“other proteins involved in innate immune response in LMH cell”, because cell supernatant cannot detect cytokine with β-actin as an internal reference.
3. Line 228: “As shown in figure. 6” should be changed to “As shown in figure. 7”.
4. Line228:“In a preliminary experiment, it was observed multiple times that the viral replication of different groups was significantly different at 72 h”, Why does the cytokine test not detect 72 hours, but 48 hours and 96 hours.
5. Line 237: “gardless of FAdV-4 GY inoculation 24 h before (Figure. 4-E)” should be changed to“….. (Figure. 7-E)”
6. Figure 3. The sample name is incorrectly labeled.
7. Figure 4. Phylogenetic tree analysis show that FAdV-HBHD isolates clustered with FAdV-E group, and serotype 4 FAdV belonged to FAdV-C group, whether the farm was infected with FAdV-4?
8. The author should improve the format of the manuscript. Line 60, Line116, Line 135, Line148, Line248, Line254-258.
9. Please check the references carefully.
10. Name the virus uniformly to avoid confusion.
The languages of the paper should be revised by the native English speaker.
Author Response

(The authors gave the same response as above.)

Reviewer 3 Report
The manuscript entitled, " FAdV-4 promotes expression of multiple cytokines and inhibits the proliferation of aHEV in LMH cells” is generally well-addressed and well-cited however; I have some comments:
Line 14: FAdV..verify abbreviation
Line 15: “………in two laying hens with severe liver injury”, is this study conducted on two laying hens? Please revise to be two flocks.
Line 38: the sentence “2016 on wards, an epidemic associated with aHEV infection has occurred in several laying hens in China”. How epidemic in several laying hens? do you means several hen farms?
Line 48: The introduction need to be revised. I suggest to add few lines about FAdV-4 coinfection and how they are related.
Line 61: please provide ethical statement
Line 62: (no. LN) what this mean? same at line 66 (No. HB). if this for Liaoning and Hebei Province? Please rewrite.
Line 69: "In total, 500 serum samples were randomly collected from individuals in the two groups.” Rewrite and add details about samples collection and handing.
Line 70: I suggest to add another subtitle for samples screening by ELISA as the title here is for samples collection or change the existing title.
Line 78: “Liver samples were obtained from dead chickens”. How many liver samples were collected?
Line 90: please provide PCR conditions that being used.
Line 95: ‘ .....liver samples of diseased chickens” and in the above one at line 77..... liver samples of infected poultry. Please ensure consistency.
Line 96: RNA was extracted using commercial kits, please specify the kits that being used. Also, is the liver samples are the same samples collected for DNA extractions or different samples?
Line 117: specify the transfection method being used in the experiment.
Line 134: the sentence “IFA was performed following the standard procedure”, add reference.
Line 137: The antibodies previously prepared in your lab, is that a reference that can be added for these antibodies.
Line 225: The sentence “co-infection with 9ave-YT also decreased the transcription ...." what is 9ave-YT? It does not mention anywhere in the text or at figure 9 as indicated. Please correct if needed.
References are repeated at the end, please revise and add DOI as possible.
Moderate editing is required.
Author Response

(The authors gave the same response as above.)

Round 2
Reviewer 3 Report
The manuscript has improved, Thank you!
I still have minor edits:
Line 124: “ the aHEV wild-type strain YT ………….and serum type 4 FAdV-I wild-type strain GY were transfected into the LMH cells” please revise to avoid confusion? is the LMH cells transfected or infected/co-infected with the viruses.
Line 294: the 2nd paragraph of discussion “We observed the co-infection of aHEV and FAdV-I in LMH cells………” need to be revised and discussed as it looks like results.
Reference # 20, need to align with all references
Author Response
Thank you for your comments. Please see attached.
